# The Thermodynamic Variational Objective

**Vaden Masrani**[1], **Tuan Anh Le**[2], **Frank Wood**[1]
[1]Department of Computer Science, University of British Columbia
[2]Department of Brain and Cognitive Sciences, MIT

## Abstract

We introduce the thermodynamic variational objective (TVO) for learning in both continuous and discrete deep generative models. The TVO arises from a key connection between variational inference and thermodynamic integration that results in a tighter lower bound to the log marginal likelihood than the standard variational evidence lower bound (ELBO) while remaining as broadly applicable. We provide a computationally efficient gradient estimator for the TVO that applies to continuous, discrete, and non-reparameterizable distributions and show that the objective functions used in variational inference, variational autoencoders, wake sleep, and inference compilation are all special cases of the TVO. We use the TVO to learn both discrete and continuous deep generative models and empirically demonstrate state of the art model and inference network learning.

## 1 Introduction

Unsupervised learning in richly structured deep latent variable models [1, 2] remains challenging. Fundamental research directions include low-variance gradient estimation for discrete and continuous latent variable models [3-7], tightening variational bounds in order to obtain better model learning [8-11], and alleviation of the associated detrimental effects on learning of inference networks [12].

We present the thermodynamic variational objective (TVO), which is based on a key connection we establish between thermodynamic integration (TI) and amortized variational inference (VI), namely that by forming a geometric path between the model and inference network, the "instantaneous ELBO" [13] that appears in VI is equivalent to the first derivative of the potential function that appears in TI [14, 15]. This allows us to formulate the log evidence as a 1D integration of the instantaneous ELBO in a unit interval, which we then approximate to form the TVO.

We demonstrate that optimizing the TVO leads to improved learning of both discrete and continuous latent-variable deep generative models. The gradient estimator we derive for optimizing the TVO has empirically lower variance than the REINFORCE [16] estimator, and unlike the reparameterization trick (which is only applicable to a limited family of continuous latent variables), applies to both continuous and discrete latent variables models.

The TVO is a lower bound to the log evidence which can be made arbitrarily tight. We empirically show that optimizing the TVO results in better inference networks than optimizing the importance weighted autoencoder (IWAE) objective [8] for which tightening of the bound is known to make inference network learning worse [12]. While this problem can be ameliorated by reducing the variance of the gradient estimator in the case of reparameterizable latent variables [17], resolving it in the case of non-reparameterizable latent variables currently involves alternating optimization of model and inference networks [18-20].

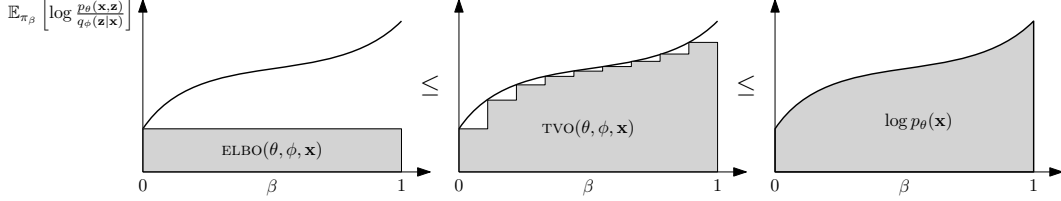

Figure 1: The thermodynamic variational objective (TVO) (center) is a finite sum numerical approximation to $\log p_\theta(\mathbf{x})$, defined by the thermodynamic variational identity (TVI) (right). The ELBO (left) is a single partition approximation of the same integral. $\pi_\beta$ is given in (7)

## 2   The Thermodynamic Variational Objective

The evidence lower bound (ELBO), used in learning variational autoencoders (VAEs), lower bounds the log evidence of a generative model $p_\theta(\mathbf{x}, \mathbf{z})$ parameterized by $\theta$ of a latent variable $\mathbf{z}$ and data $\mathbf{x}$. It can be written as the log evidence minus a Kullback-Leibler (KL) divergence

$$\text{ELBO}(\theta, \phi, \mathbf{x}) := \log p_\theta(\mathbf{x}) - \text{KL}\left(q_\phi(\mathbf{z}|\mathbf{x})||p_\theta(\mathbf{z}|\mathbf{x})\right), \tag{1}$$

where $q_\phi(\mathbf{z}|\mathbf{x})$ is an inference network parameterized by $\phi$. As illustrated in Figure 1, the TVO

$$\underbrace{\frac{1}{K}\left[\text{ELBO}(\theta, \phi, \mathbf{x}) + \sum_{k=1}^{K-1}\mathbb{E}_{\pi_{\beta_k}}\left[\log\frac{p_\theta(\mathbf{x}, \mathbf{z})}{q_\phi(\mathbf{z}\,|\,\mathbf{x})}\right]\right]}_{\text{TVO}(\theta, \phi, \mathbf{x})} \leq \underbrace{\int_0^1 \mathbb{E}_{\pi_\beta}\left[\log\frac{p_\theta(\mathbf{x}, \mathbf{z})}{q_\phi(\mathbf{z}\,|\,\mathbf{x})}\right]d\beta = \log p_\theta(\mathbf{x})}_{\text{THERMODYNAMIC VARIATIONAL IDENTITY}} \tag{2}$$

lower bounds the log evidence by using a Riemann sum approximation to the TVI, a one-dimensional integral over a scalar $\beta$ in a unit interval which evaluates to the log model evidence $\log p_\theta(\mathbf{x})$.

The integrand, which is a function of $\beta$, is an expectation of the so-called "instantaneous ELBO" [13] under $\pi_\beta(\mathbf{z})$, a geometric combination of $p_\theta(\mathbf{x}, \mathbf{z})$ and $q_\phi(\mathbf{z}|\mathbf{x})$ which we formally define in §3. Remarkably, at $\beta = 0$, the integrand equals the ELBO. This therefore allows us to view the ELBO as a single-term left Riemann sum of the TVI. At $\beta = 1$, the integrand equals to the evidence upper bound (EUBO). This sheds a new unifying perspective on the VAE and wake-sleep objectives, which we explore in detail in §5 and Appendix G.

## 3   Connecting Thermodynamic Integration and Variational Inference

Suppose there are two unnormalized densities $\tilde{\pi}_i(\mathbf{z})$ $(i = 0, 1)$ and corresponding normalizing constants $Z_i := \int \tilde{\pi}_i(\mathbf{z})\,d\mathbf{z}$, which together define the normalized densities $\pi_i(\mathbf{z}) := \tilde{\pi}_i(\mathbf{z})/Z_i$. We can typically evaluate the unnormalized densities but cannot evaluate the normalizing constants.

While calculating the normalizing constants individually is usually intractable, thermodynamic integration [14, 15] allows us to compute the log of the ratio of the normalizing constants, $\log Z_1/Z_0$. To do so, we first form a family of unnormalized densities (or a "path") parameterized by $\beta \in [0, 1]$ between the two distributions of interest

$$\tilde{\pi}_\beta(\mathbf{z}) := \tilde{\pi}_1(\mathbf{z})^\beta \tilde{\pi}_0(\mathbf{z})^{1-\beta} \tag{3}$$

with the corresponding normalizing constants and normalized densities

$$Z_\beta := \int \tilde{\pi}_\beta(\mathbf{z})\,d\mathbf{z}, \quad \text{and} \quad \pi_\beta(\mathbf{z}) := \tilde{\pi}_\beta(\mathbf{z})/Z_\beta. \tag{4}$$

Following Neal [15], we will find it useful to define a potential energy function $U_\beta(\mathbf{z}) := \log \tilde{\pi}_\beta(\mathbf{z})$ along with its first derivative $U'_\beta(\mathbf{z}) = \frac{dU_\beta(\mathbf{z})}{d\beta}$. We can then estimate the log of the ratio of the normalizing constants via the identity central to TI, derived in Appendix A,

$$\log Z_1 - \log Z_0 = \int_0^1 \mathbb{E}_{\pi_\beta}\left[U'_\beta(\mathbf{z})\right]d\beta. \tag{5}$$

Our key insight connecting TI and VI is the following. If we set

$$\tilde{\pi}_0(\mathbf{z}) := q_\phi(\mathbf{z} \,|\, \mathbf{x}) \quad Z_0 = \int q_\phi(\mathbf{z} \,|\, \mathbf{x})\, \mathrm{d}\mathbf{z} = 1$$

$$\tilde{\pi}_1(\mathbf{z}) := p_\theta(\mathbf{x}, \mathbf{z}) \quad Z_1 = \int p_\theta(\mathbf{x}, \mathbf{z})\, \mathrm{d}\mathbf{z} = p_\theta(\mathbf{x}) \tag{6}$$

this results in a geometric path between the variational distribution $q_\phi(\mathbf{z}|\mathbf{x})$ and the model $p_\theta(\mathbf{x}, \mathbf{z})$

$$\tilde{\pi}_\beta(\mathbf{z}) := p_\theta(\mathbf{x}, \mathbf{z})^\beta q_\phi(\mathbf{z}|\mathbf{x})^{1-\beta} \quad \text{and} \quad \pi_\beta(\mathbf{z}) := \frac{\tilde{\pi}_\beta(\mathbf{z})}{Z_\beta}, \tag{7}$$

where the first derivative of the potential is equal to the "instantaneous ELBO" [13]

$$U'_\beta(\mathbf{z}) = \log \frac{p_\theta(\mathbf{x}, \mathbf{z})}{q_\phi(\mathbf{z}|\mathbf{x})}. \tag{8}$$

Substituting (8) and $Z_0 = 1$ and $Z_1 = p_\theta(\mathbf{x})$ into (5) results in the *thermodynamic variational identity*

$$\log p_\theta(\mathbf{x}) = \int_0^1 \mathbb{E}_{\pi_\beta} \left[ \log \frac{p_\theta(\mathbf{x}, \mathbf{z})}{q_\phi(\mathbf{z} \,|\, \mathbf{x})} \right] d\beta. \tag{9}$$

This means that $\log p_\theta(\mathbf{x})$ can be expressed as a one-dimensional integral of an expectation of the instantaneous ELBO under $\pi_\beta$ from $\beta = 0$ to $\beta = 1$ (see Figure 1 (right)).

To obtain the thermodynamic variational objective (TVO) defined in (2), we lower bound the integral in (9) using a left Riemann sum. That this is in fact a lower bound follows from observation that the integrand is monotonically increasing, as shown in Appendix B. This is a result of our choice of path in (7), which allows us to show the derivative of the integrand is equal to the variance of $U'_\beta(\mathbf{z})$ under $\pi_\beta(\mathbf{z})$ and is therefore non-negative. For equal spacing of the partitions, where $\beta_k = k/K$, we arrive at the TVO in (2), illustrated in Figure 1 (middle). We present a generalized variant with non-equal spacing in Appendix C.

Maximizing the ELBO$(\theta, \phi, \mathbf{x})$ can be seen as a special case of the TVO, since for $\beta = 0$, $\pi_\beta(\mathbf{z}) = q_\phi(\mathbf{z}|\mathbf{x})$, and so the integrand in (9) becomes $\mathbb{E}_{q_\phi(\mathbf{z}|\mathbf{x})} \left[ \log \frac{p_\theta(\mathbf{x}, \mathbf{z})}{q_\phi(\mathbf{z} \,|\, \mathbf{x})} \right]$, which is equivalent to the definition of ELBO in (1). Because the integrand is increasing, we have

$$\text{ELBO}(\theta, \phi, \mathbf{x}) \leq \text{TVO}(\theta, \phi, \mathbf{x}) \leq \log p_\theta(\mathbf{x}), \tag{10}$$

which means that the TVO is an alternative to IWAE for tightening the variational bounds. In Appendix D we show maximizing the TVO is equivalent to minimizing a divergence between the variational distribution and the true posterior $p_\theta(\mathbf{z} \,|\, \mathbf{x})$.

The integrand in (9) is typically estimated by long running Markov chain Monte Carlo chains computed at different values of $\pi_\beta(\mathbf{z})$ [21, 22]. Instead, we propose a simple importance sampling mechanism that allows us to reuse samples across an arbitrary number of discretizations and which is compatible with gradient-based learning.

## 4 Optimizing the TVO

We now provide a novel score-function based gradient estimator for the TVO which does not require the reparameterization trick.

**Gradients** To use the TVO as a variational objective we must be able to differentiate through terms of the form $\nabla_\lambda \mathbb{E}_{\pi_{\lambda,\beta}}[f_\lambda(\mathbf{z})]$, where both $\pi_{\lambda,\beta}(\mathbf{z})$ and $f_\lambda(\mathbf{z})$ are parameterized by $\lambda$, and $\pi_{\lambda,\beta}(\mathbf{z})$ contains an intractable normalizing constant. In the TVO, $f_\lambda(\mathbf{z})$ is the instantaneous ELBO and $\lambda := \{\theta, \phi\}$, but our method is applicable for generic $f_\lambda(\mathbf{z}) : \mathbb{R}^M \mapsto \mathbb{R}$.

We can compute such terms using the *covariance gradient estimator* (derived in Appendix E)

$$\nabla_\lambda \mathbb{E}_{\pi_{\lambda,\beta}}[f_\lambda(\mathbf{z})] = \mathbb{E}_{\pi_{\lambda,\beta}}[\nabla_\lambda f_\lambda(\mathbf{z})] + \text{Cov}_{\pi_{\lambda,\beta}}[\nabla_\lambda \log \tilde{\pi}_{\lambda,\beta}(\mathbf{z}), f_\lambda(\mathbf{z})] \tag{11}$$

We emphasize that, like REINFORCE, our estimator relies on the log-derivative trick, but crucially *unlike* REINFORCE, doesn't require differentiating through the normalizing constant $Z_\beta = \int \tilde{\pi}_{\lambda,\beta}(\mathbf{z}) \, d\mathbf{z}$. We clarify the relationship between our estimator and REINFORCE in Appendix F.

The covariance in (11) has the same dimensionality as $\lambda \in \mathbb{R}^D$ because it is between $\nabla_\lambda \log \tilde{\pi}_{\lambda,\beta}(\mathbf{z}) \in \mathbb{R}^D$ and $f_\lambda(\mathbf{z}) \in \mathbb{R}$ and is defined as

$$\text{Cov}_{\pi_{\lambda,\beta}}(\mathbf{a}, b) := \mathbb{E}_{\pi_{\lambda,\beta}} \left[ (\mathbf{a} - \mathbb{E}_{\pi_{\lambda,\beta}}[\mathbf{a}])(b - \mathbb{E}_{\pi_{\lambda,\beta}}[b]) \right]. \tag{12}$$

To estimate this, we first estimate the inner expectations which are then used in estimating the outer expectation. Thus, estimating the gradient in (11) requires estimating expectations under $\pi_\beta$.

**Expectations**   By using $q_\phi(\mathbf{z}|\mathbf{x})$ as the proposal distribution in $S$-sample importance sampling, we can estimate an expectation of a general function $f(\mathbf{z})$ under any $\pi_\beta(\mathbf{z})$ by simply raising each unnormalized importance weight to the power $\beta$ and normalizing:

$$\mathbb{E}_{\pi_\beta}[f(\mathbf{z})] \approx \sum_{s=1}^{S} \overline{w_s^\beta} f(\mathbf{z}_s), \tag{13}$$

where $\mathbf{z}_s \sim q_\phi(\mathbf{z}|\mathbf{x})$, $\overline{w_s^\beta} := w_s^\beta / \sum_{s'=1}^{S} w_{s'}^\beta$ and $w_s := \frac{p_\theta(\mathbf{x}, \mathbf{z}_s)}{q_\phi(\mathbf{z}_s|\mathbf{x})}$. This follows because each unnormalized importance weight can be expressed as

$$\frac{\tilde{\pi}_\beta(\mathbf{x}, \mathbf{z}_s)}{q_\phi(\mathbf{z}_s|\mathbf{x})} = \frac{p_\theta(\mathbf{x}, \mathbf{z}_s)^\beta q_\phi(\mathbf{z}_s|\mathbf{x})^{1-\beta}}{q_\phi(\mathbf{z}_s|\mathbf{x})} = \frac{p_\theta(\mathbf{x}, \mathbf{z}_s)^\beta}{q_\phi(\mathbf{z}_s|\mathbf{x})^\beta} = \left( \frac{p_\theta(\mathbf{x}, \mathbf{z}_s)}{q_\phi(\mathbf{z}_s|\mathbf{x})} \right)^\beta = w_s^\beta. \tag{14}$$

Instead of sampling $SK$ times, we can reuse $S$ samples $\mathbf{z}_s \sim q_\phi(\mathbf{z}|\mathbf{x})$ across an arbitrary number of terms, since evaluating the normalized weight $\overline{w_s^{\beta_k}}$ only requires raising each weight to different powers of $\beta_k$ before normalizing. Reusing samples in this way is a use of the method known as "common random numbers" and we include experimental results showing it reduces the variance of the covariance estimator in Appendix F [23].

The covariance estimator does not require $\mathbf{z}$ to be reparameterizable, which means it can be used in the cases of both non-reparameterizable continuous latent variables and discrete latent variables (without modifying the model using continuous relaxations [24, 25]).

# 5   Generalizing Variational Objectives

As previously observed, the left single Riemann approximation of the TVI equals the ELBO, while the right endpoint ($\beta = 1$) is equal to the EUBO. The EUBO is analogous to the ELBO but under the true posterior and is defined

$$\text{EUBO}(\theta, \phi, \mathbf{x}) := \mathbb{E}_{p_\theta(\mathbf{z}|\mathbf{x})} \left[ \log \frac{p_\theta(\mathbf{x}, \mathbf{z})}{q_\phi(\mathbf{z}|\mathbf{x})} \right]. \tag{15}$$

We also have the following identity

$$\text{EUBO}(\mathbf{x}, \theta, \phi) = \log p_\theta(\mathbf{x}) + \text{KL}\left( p_\theta(\mathbf{z}|\mathbf{x}) || q_\phi(\mathbf{z}|\mathbf{x}) \right) \tag{16}$$

which should be contrasted against (1). We define an upper-bound variant of the TVO using the right (rather than left) Riemann sum. Setting $\beta_k = k/K$

$$\text{TVO}_K^U(\theta, \phi, \mathbf{x}) := \frac{1}{K} \left[ \text{EUBO}(\theta, \phi, \mathbf{x}) + \sum_{k=1}^{K-1} \mathbb{E}_{\pi_{\beta_k}} \left[ \log \frac{p_\theta(\mathbf{x}, \mathbf{z})}{q_\phi(\mathbf{z}|\mathbf{x})} \right] \right] \geq \log p(\mathbf{x}). \tag{17}$$

The wake-sleep (WS) [18] and reweighted wake-sleep (RWS) [19] algorithms have traditionally been viewed as using different objectives during the wake and sleep phase. The endpoints of the TVI, which the TVO approximates, correspond to the two objectives used in wake-sleep. We can therefore view WS as alternating between between $\text{TVO}_1^L$ and $\text{TVO}_1^U$, i.e. a left and right single term Riemann approximation to the TVI. We show this algebraically in Appendix G and additionally, show how the objectives used in variational inference [26], variational autoencoders [1, 2], and inference compilation [27] are all special cases of $\text{TVO}_1^L$ and $\text{TVO}_1^U$. We refer the reader to [20] for a further discussion of the wake-sleep algorithm.

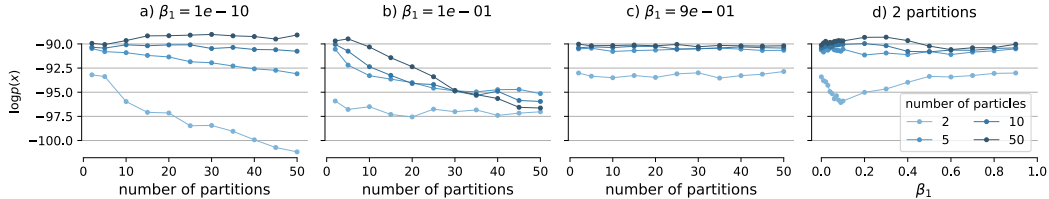

Figure 2: Investigation of how number of particles $S$, number of partitions $K$, and $\beta_1$ affect learning of the generative model. In the first three plots (a-c), we vary $S$ and $K$ for different values of $\beta_1$ and observe that while $S$ should be as high as possible, there is an optimal value for $K$, beyond which performance begins to degrade. Assuming $\beta_1$ is well-chosen, we see that as few as $K = 2$ partitions can result in good model learning, as seen in the last plot (d).

## 6    Related Work

Thermodynamic integration was originally developed in physics to calculate the difference in free energy of two molecular systems [28]. Neal [15] and Gelman and Meng [14] then introduced TI into the statistics community to calculate the ratios of normalizing constants of general probability models. TI is now commonly used in phylogenetics to calculate the Bayes factor $B = p(x|M_1)/p(x|M_0)$, where $M_0, M_1$ are two models specifying (for instance) tree topologies and branch lengths [22, 29, 30]. We took inspiration from Fan et al. [31] who replaced the "power posterior" $p(\theta|\mathbf{x}, M, \beta) = p(x|\theta, M)^\beta p(\theta, M)/Z_\beta$ of Xie et al. [29] with $p(\theta|\mathbf{x}, M, \beta) = [p(\mathbf{x}|\theta, M)p(\theta|M)]^\beta [p_0(\theta|M)]^{1-\beta}/Z_\beta$, where $p_0(\theta|M)$ is a tractable reference distribution chosen to facilitate sampling. That the integrand in (9) is strictly increasing was observed by Lartillot and Philippe [22].

We refer the reader to Titsias and Ruiz [32] for a summary of the numerous advances in variational methods over recent years. The method most similar to our own was proposed by Bornschein et al. [33], who introduced another way of improving deep generative modeling through geometrically interpolating between distributions and using importance sampling to estimate gradients. Unlike the TVO, they define a lower bound on the marginal likelihood of a modified model defined as $(p_\theta(\mathbf{x}, \mathbf{z})q_\phi(\mathbf{z}|\mathbf{x})q(\mathbf{x}))^{1/2}/Z$ where $q(\mathbf{x})$ is an auxiliary distribution.

Grosse et al. [34] studied annealed importance sampling (AIS), a related technique that estimates partition functions using a sequence of intermediate distributions to form a product of ratios of importance weights. They observe the geometric path taken in AIS is equivalent to minimizing a weighted sum of KL divergences, and use this insight to motivate an alternative path. To the best of our knowledge, our work is the first to explicitly connect TI and VI.

## 7    Experiments

### 7.1    Discrete Deep Generative Models

We use the TVO to learn the parameters of a deep generative model with discrete latent variables.[1] We use the binarized MNIST dataset with the standard train/validation/test split of 50k/10k/10k [35]. We train a sigmoid belief network, described in detail in Appendix I, using the TVO with the Adam optimizer. In the first set of experiments we investigate the effect of the discretization $\beta_{0:K}$, number of partitions $K$ and number of particles $S$. We then compare against variational inference for Monte Carlo objectives (VIMCO) and RWS (with the wake-$\phi$ objective) state-of-the-art IWAE-based methods for learning discrete latent variable models [20]. All figures have been smoothed for clarity.

**The effect of $S$, $K$, and $\beta$ locations** We expect that increasing the number of partitions $K$ makes the Riemann sum approximate the integral over $\beta$ more tightly. However, with each addition term, we add noise due to the use of importance sampling to estimate the expectation $\mathbb{E}_{\pi_\beta}[\log p/q]$. Importance sampling estimates of points on the curve further to the right are likely to be more biased because $\pi_\beta$ gets further from $q$ as we increase $\beta$. We found the combination of these two effects means that there

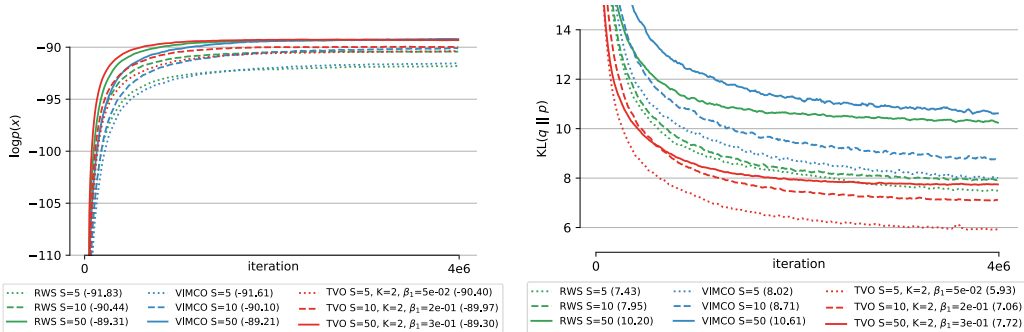

Figure 3: Comparisons with baselines on a held out test set. *(Left)* Learning curves for different methods. For TVO outperforms other methods both in terms of speed of convergence and the learned model for $S < 50$. At $S = 50$ VIMCO achieves a higher test log evidence but takes longer to converge than the TVO. *(Right)* KL divergence between current $q$ and $p$ (which measures how well $q$ "tracks" $p$) is lowest for TVO.

is a "sweet spot," or an optimal number of partitions beyond which adding more partitions becomes detrimental to performance.

We have empirically observed that the curve in Figure 1 is often rising sharply from $\beta = 0$ until a point of maximum curvature $\beta^*$, after which it is almost flat until $\beta = 1$, as seen in Figure 4. We hypothesized that if $\beta_1$ is located far before $\beta^*$ (the point of maximum curvature), a large number of additional partitions would be needed to capture additional area, while if $\beta_1$ is located after $\beta^*$, additional partitions would simply incur a high cost of bias without significantly tightening the bound. To investigate this, we choose small ($10^{-10}$), medium (0.1) and large (0.9) values of $\beta_1$, and logarithmically space the remaining $\beta_{2:K}$ between $\beta_1$ and 1. For each value of $\beta_1$ we train the discrete generative model for $K \in \{2, 5, 10, \dots, 50\}$ and $S \in \{2, 5, 10, 50\}$, and show the test log evidence at the last iteration of each trial, approximated by evaluating the IWAE loss with 5000 samples.

Our hypothesis is corroborated in Figure 2, where we observe in Figure 2a that for $\beta_1 = 10^{-10}$ a large number of partitions are needed to approximate the integral. In Figure 2b we increase $\beta_1$ to $10^{-1}$ and observe only a few partitions are needed to improve performance, after which adding additional partitions becomes detrimental to model learning.

From Figure 2c we can see that if $\beta_1$ is chosen to be well beyond $\beta^*$, the Riemann sum cannot recover the "lost" area even if the number of partitions is increased. That the performance does not degrade in this case is due to the fact that for sufficiently high $\beta_k$, the curve in Figure 1 is flat and therefore $\pi_{\beta_k} \approx \pi_{\beta_k+1} \approx p_\theta(\mathbf{z} \mid \mathbf{x})$. We also observe that increasing number of samples $S$—which decreases importance sampling bias per partition—improves performance in all cases.

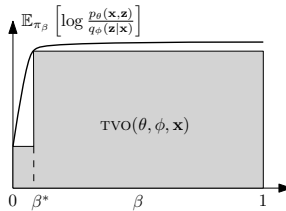

Figure 4: The location of $\beta^*$, the point of maximum curvature.

In our second experiment, shown in the Figure 2d, we fix $K = 2$ and investigate the quality of the learned generative model for different $\beta_1$, This plot clearly shows $\beta^*$ is somewhere near 0.3, as model learning improves as $\beta_1$ approaches this point then begins to degrade.

Given these results, we recommend using as many particles $S$ as possible and performing a hyperparameter search over $\beta_1$ (with $K = 2$) when using the TVO objective. We leave finding the optimal placement of discretization points to future work.

**Performance** In Figure 3 (left), we compare the TVO against VIMCO and RWS with the wake-$\phi$ objective, the state-of-the-art IWAE-based methods for learning discrete latent variable models [20]. For $S < 50$, the TVO outperforms both methods in terms of speed of convergence and the final test log evidence $\log p_\theta(\mathbf{x})$, estimated using 5000 IWAE particles as before. At $S = 50$ VIMCO achieves a higher test log evidence but converges more slowly.

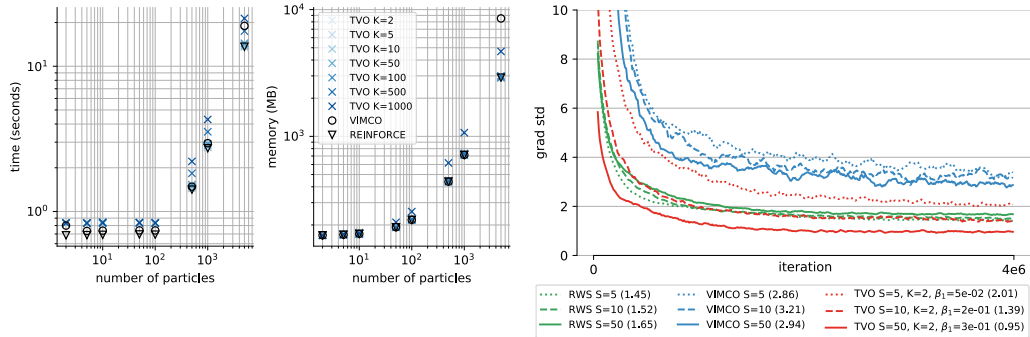

Figure 5: Computational and gradient estimator efficiency. *(Left)* Time and memory efficiency of the TVO with increasing number of partitions vs baselines, measured for 100 iterations of optimization. Increasing the number of partitions is much cheaper than increasing the number of particles. *(Right)* Standard deviation of the gradient estimator for each objective. TVO is lowest variance, VIMCO is highest variance, RWS is in the middle.

We also investigate the quality of the learned inference network by plotting the KL divergence (averaged over the test set) between the current $q$ and current $p$ as training progresses (Figure 3 (right)). This indicates how well $q$ "tracks" $p$. This is estimated as log evidence minus ELBO where the former is estimated as before and the latter is estimated using 5000 Monte Carlo samples. The KL is lowest for TVO.

Somewhat surprisingly, for all methods, increasing number of particles makes the KL worse. We speculate that this is due to the "tighter bounds" effect of Rainforth et al. [12], who showed that increasing the number of samples can positively affect model learning but adversely affect inference network learning, thereby increasing the KL between the two.

**Efficiency** Since we use $K = 2$ partitions for the same number of particles $S$, the time and memory complexity of TVO is double that of other methods. While this is true, in both time and memory cases, the constant factor for increasing $S$ is much higher than for increasing $K$. As shown in Figure 5 (left), it is virtually free to increase number of partitions. This is because for each new particle, we must additionally sample from the inference network and score the sample under both $p$ and $q$ to obtain a weight. On the other hand, we can reuse the $S$ samples and corresponding weights in estimating values for the $K + 1$ terms in the Riemann sum. Thus, the region of the the computation graph that is dependent on $K$ is *after* the expensive sampling and scoring, and only involves performing basic operations on additional matrices of size $S \times K$.

**Variance** In Figure 5 (right), we plot the standard deviation of the gradient estimator for each method, where we compute the standard deviation for the $d^{\text{th}}$ element of the gradient estimated over 10 samples and take the average across all $D$.

The gradient estimator of the TVO has lower variance than both VIMCO, which uses REINFORCE with a control variate as a gradient estimator and RWS which can calculate the gradient without reparameterizing or using the log-derivative trick. At $S = 5$, RWS has lower gradient variance but its performance is worse in terms of both model and inference learning.

## 7.2 Continuous Deep Generative Models

Using the binarized MNIST dataset and experimental design described above, we also evaluated our method on a deep generative model with continuous latent variables. The model is described in detail in Appendix I. For each $S \in \{5, 10, 50\}$ we sweep over $K \in \{2, ..., 6\}$ and 20 $\beta_1$ values linearly spaced between $10^{-2}$ and 0.9. We optimize the objectives using the Adam optimizer with default parameters.

**Performance** In Figure 6 (left), we train the model using the TVO and compare against the same model trained using the single sample VAE objective and multisample IWAE objective. The TVO outperforms the VAE and performs competitively with IWAE at 50 samples, despite not using the reparameterization trick. IWAE is the top performing objective in all cases. As in the discrete case,

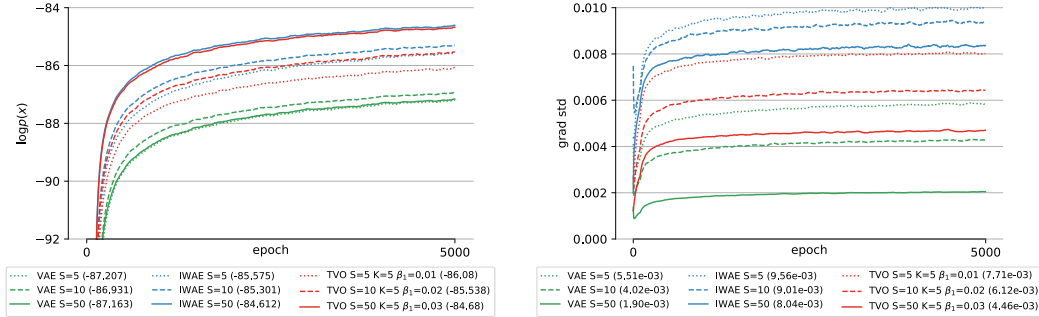

Figure 6: Learning curves for learning continuous deep generative models using different objectives. *(Left)* Despite not using the reparameterization trick, TVO outperforms VAEs and is competitive with IWAE at 50 samples. For all S, IWAE $>$ TVO $>$ VAE. *(Right)* Standard deviation of the gradient estimator for each objective. The TVO has lower variance than IWAE but higher than VAE.

increasing the number of particles $S$ improves model learning for all methods, but the improvement is most significant for the TVO. Interestingly VAE performance actually *decreases* when the number of samples increases from 10 to 50. A similar effect was noticed by Burda et al. [8] on the omniglot dataset.

**Variance** In Figure 6 (right), we plot the standard deviation of each method's gradient estimator. The standard deviation of the TVO estimator falls squarely between VAE (best) and IWAE (worst). The variance of each method improves as the number of samples increases, and as in the discrete model, the improvement is most significant in the case of TVO. Unlike in the discrete case, the variance does not decrease as the optimization proceeds, but plateaus early and then gradually increases. In Appendix F we include additional experiments to evaluate the properties of the covariance gradient estimator when used on the ELBO.

For both IWAE and the TVO, increasing the number of samples leads to decreased gradient variance and improved model learning. However, IWAE has the best performance but the highest variance across the three models. These results lend support to the conclusions of Rainforth et al. [12] who observe that the variance of a gradient estimator "is not always a good barometer for the effectiveness of a gradient estimation scheme."

## 8    Conclusions

The thermodynamic variational objective represents a new way to tighten evidence bounds and is based on a tight connection between variational inference and thermodynamic integration. We demonstrated that optimizing the TVO can have a positive impact on the learning of discrete deep generative models and can perform as well as using the reparameterization trick to learn continuous deep generative models.

The weakness of our method lies in choosing the discretization points. This does, however, point out opportunities for future work wherein we adaptively select optimal positions of the $\beta_{1:K}$ points, perhaps using techniques from the Bayesian numerical quadrature literature [36–38].

The approximate path integration perspective provided by our development of the TVO also sheds light on the connection between otherwise disparate deep generative model learning techniques. In particular, the TVO integration perspective points to ways to improve wake-sleep via tightening the EUBO using similar integral upper-bounding techniques. Further experimentation is warranted to explore how TVO insights can be applied to all special cases of the TVO including non-amortized variational inference and to the use of the TVO as a compliment to annealing importance sampling for final model evidence evaluation.

**Acknowledgments**

We would like to thank Trevor Campbell, Adam Ścibior, Boyan Beronov, and Saifuddin Syed for their helpful comments on early drafts of this manuscript. Tuan Anh Le's research leading to these results is supported by EPSRC DTA and Google (project code DF6700) studentships. We acknowledge the support of the Natural Sciences and Engineering Research Council of Canada (NSERC), the Canada CIFAR AI Chairs Program, Compute Canada, Intel, and DARPA under its D3M and LWLL programs.

## Footnotes

[1]Code to reproduce all experiments is available at: https://github.com/vmasrani/tvo.

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
