[Supplementary Material · supp.pdf]

# A Thermodynamic Integration

Thermodynamic integration is a technique used in physics and phylogenetics to approximate intractable normalized constants of high dimensional distributions [14, 15]. It is based on the observation that it is easier to calculate the ratio of two unknown normalizing constants than it is to calculate the constants themselves. More formally, consider two densities over space $\mathcal{Z}$

$$\pi_i(\mathbf{z}) = \frac{\tilde{\pi}_i(\mathbf{z})}{Z_i}, \qquad Z_i = \int_{\mathcal{Z}} \tilde{\pi}(\mathbf{z}) \, d\mathbf{z}, \qquad i \in \{0, 1\}. \tag{18}$$

To apply TI, we form a continuous family (or "path") between $\pi_0(\mathbf{z})$ and $\pi_1(\mathbf{z})$ via a scalar parameter $\beta \in [0, 1]$

$$\pi_\beta(\mathbf{z}) = \frac{\tilde{\pi}_\beta(\mathbf{z})}{Z_\beta} = \frac{\tilde{\pi}_1(\mathbf{z})^\beta \tilde{\pi}_0(\mathbf{z})^{1-\beta}}{Z_\beta}, \qquad Z_\beta = \int_{\mathcal{Z}} \tilde{\pi}_\beta(\mathbf{z}) \, d\mathbf{z}, \qquad \beta \in [0, 1]. \tag{19}$$

The central identity that allows us to compute the ratio $\log(Z_1/Z_0)$ is derived as follows. Assuming we can exchange integration with differentiation,

$$\begin{aligned}
\frac{\partial \log Z_\beta}{\partial \beta} &= \frac{1}{Z_\beta} \frac{\partial}{\partial \beta} Z_\beta \\
&= \frac{1}{Z_\beta} \frac{\partial}{\partial \beta} \int \tilde{\pi}_\beta(\mathbf{z}) \, d\mathbf{z} \\
&= \int \frac{1}{Z_\beta} \frac{\partial}{\partial \beta} \tilde{\pi}_\beta(\mathbf{z}) \, d\mathbf{z} \\
&= \int \frac{\tilde{\pi}_\beta(\mathbf{z})}{Z_\beta} \frac{\partial}{\partial \beta} \log \tilde{\pi}_\beta(\mathbf{z}) \, d\mathbf{z},
\end{aligned}$$

which directly implies

$$\frac{\partial \log Z_\beta}{\partial \beta} = \mathbb{E}_{\pi_\beta} \left[ U'_\beta(\mathbf{z}) \right], \tag{20}$$

where the quantity $U_\beta(\mathbf{z}) = \log \tilde{\pi}_\beta(\mathbf{z})$ is referred to as the "potential" in statistical physics and $U'_\beta(\mathbf{z}) := \frac{\partial}{\partial \beta} U_\beta(\mathbf{z})$. The variable $\beta$ can be interpreted as the inverse temperature parameter. Because one can typically compute $\log \tilde{\pi}_\beta(\mathbf{z})$, (20) allows us to exchange the first derivative of something we cannot compute with an expectation over something we can compute. Then, to calculate the ratio $\log(Z_1/Z_0)$ we integrate out $\beta$ on both sides of (20)

$$\int_0^1 \frac{\partial \log Z_\beta}{\partial \beta} d\beta = \int_0^1 \mathbb{E}_{\pi_\beta} \left[ U'_\beta(\mathbf{z}) \right] d\beta \tag{21}$$

which via the fundamental theorem of calculus results in

$$\log(Z_1) - \log(Z_0) = \int_0^1 \mathbb{E}_{\pi_\beta} \left[ U'_\beta(\mathbf{z}) \right] d\beta. \tag{22}$$

# B The Increasing Integrand

## B.1 Notation

$$\log p_\theta(\mathbf{x}) = \int_0^1 g(\beta) d\beta \tag{23}$$

$$g(\beta) = \mathbb{E}_{\pi_\beta(\mathbf{z})} \left[ U'(\mathbf{z}) \right] \tag{24}$$

$$U'(\mathbf{z}) = \log \frac{p_\theta(\mathbf{x}, \mathbf{z})}{q_\phi(\mathbf{z} \mid \mathbf{x})} \tag{25}$$

Given our choice of geometric path $\pi_\beta(\mathbf{z}) = \tilde{\pi}_\beta(\mathbf{z})/Z_\beta$, $\tilde{\pi}_\beta(\mathbf{z}) = p(\mathbf{x}, \mathbf{z})^\beta q(\mathbf{z} \mid \mathbf{x})^{1-\beta}$, the potential $U'(\mathbf{z}) = \frac{\partial}{\partial \beta} \log \tilde{\pi}_\beta(\mathbf{z})$ loses its dependency on $\beta$ after differentiating. This allows us to show

$$\frac{\partial}{\partial \beta} g(\beta) = \text{Var}_{\pi_\beta(\mathbf{z})} [U'(\mathbf{z})] \tag{26}$$

which means $\frac{\partial}{\partial\beta}g(\beta) \geq 0, \forall \beta \in [0,1]$ and therefore that $g(\beta)$ is monotonically non-decreasing. Changes between lines are tracked in blue.

*Proof of Equation* (26).

$$
\begin{aligned}
\frac{\partial}{\partial\beta}g(\beta) &= \frac{\partial}{\partial\beta}\,\mathbb{E}_{\pi_\beta(\mathbf{z})}\left[U'(\mathbf{z})\right] \\
&= \frac{\partial}{\partial\beta}\left[\int \pi_\beta(\mathbf{z})U'(\mathbf{z})\,\mathrm{d}\mathbf{z}\right] \\
&= \int U'(\mathbf{z})\frac{\partial}{\partial\beta}\pi_\beta(\mathbf{z})\,\mathrm{d}\mathbf{z} \\
&= \int U'(\mathbf{z})\frac{\partial}{\partial\beta}\left[Z_\beta^{-1}\tilde{\pi}_\beta(\mathbf{z})\right]\,\mathrm{d}\mathbf{z} \\
&= \int U'(\mathbf{z})\left[\tilde{\pi}_\beta(\mathbf{z})\frac{\partial}{\partial\beta}Z_\beta^{-1} + Z_\beta^{-1}\frac{\partial}{\partial\beta}\tilde{\pi}_\beta(\mathbf{z})\right]\mathrm{d}\mathbf{z}\,.
\end{aligned}
$$

Now we use the "inverse log-derivative" trick $\frac{\partial}{\partial x}(f(x)^{-1}) = -\frac{1}{f(x)}\frac{\partial}{\partial x}\log f(x)$ on the first term, and the log-derivative trick on the second

$$
= \int U'(\mathbf{z})\left[\tilde{\pi}_\beta(\mathbf{z})\frac{-1}{Z_\beta}\frac{\partial}{\partial\beta}\log Z_\beta + \frac{1}{Z_\beta}\tilde{\pi}_\beta(\mathbf{z})\frac{\partial}{\partial\beta}\log\tilde{\pi}_\beta(\mathbf{z})\right]\mathrm{d}\mathbf{z} \tag{27}
$$

$$
= \int U'(\mathbf{z})\left[-\pi_\beta(\mathbf{z})\frac{\partial}{\partial\beta}\log Z_\beta + \pi_\beta(\mathbf{z})\frac{\partial}{\partial\beta}\log\tilde{\pi}_\beta(\mathbf{z})\right]\mathrm{d}\mathbf{z}, \tag{28}
$$

Then we use (20) on the first term, and the definition of $U'(\mathbf{z})$ in the second

$$
= \int U'(\mathbf{z})\left[-\pi_\beta(\mathbf{z})\mathbb{E}_{\pi_\beta(\mathbf{z})}\left[U'(\mathbf{z})\right] + \pi_\beta(\mathbf{z})U'(\mathbf{z})\right]\mathrm{d}\mathbf{z} \tag{29}
$$

$$
= -\int \pi_\beta(\mathbf{z})U'(\mathbf{z})\,\mathbb{E}_{\pi_\beta(\mathbf{z})}\left[U'(\mathbf{z})\right]\mathrm{d}\mathbf{z} + \int U'(\mathbf{z})U'(\mathbf{z})\pi_\beta(\mathbf{z})\mathrm{d}\mathbf{z} \tag{30}
$$

Finally we rearrange, noting that the expectation is a scalar and can therefore come out of the integrand

$$
= -\left[\mathbb{E}_{\pi_\beta(\mathbf{z})}\left[U'(\mathbf{z})\right]\right]\left[\int \pi_\beta(\mathbf{z})U'(\mathbf{z})\,\mathrm{d}\mathbf{z}\right] + \int U'(\mathbf{z})U'(\mathbf{z})\pi_\beta(\mathbf{z})\,\mathrm{d}\mathbf{z} \tag{31}
$$

$$
= -\left[\mathbb{E}_{\pi_\beta(\mathbf{z})}[U'(\mathbf{z})]\right]^2 + \mathbb{E}_{\pi_\beta(\mathbf{z})}\left[U'(\mathbf{z})^2\right] \tag{32}
$$

$$
= \mathrm{Var}_{\pi_\beta(\mathbf{z})}[U'(\mathbf{z})]. \tag{33}
$$

Therefore,

$$
\frac{\partial}{\partial\beta}g(\beta) = \mathrm{Var}_{\pi_\beta(\mathbf{z})}[U'(\mathbf{z})]. \tag{34}
$$

$\square$

## C   The generalized TVO

The TVO presented in §2 is a lower bound to $\log p_\theta(\mathbf{x})$ using a left Riemann sum approximation to the thermodynamic variational identity. Using the right Riemann sum results in an upper bound which can be minimized (rather than maximized) during optimization (cf. §5). This loss is used in the inference compilation and during the sleep-phase $\phi$ update in the Wake-Sleep algorithm. Below we present both the upper-bound and lower-bound variants of the TVO, with non-equally spaced partitions $0 = \beta_0 < \beta_1 < \cdots < \beta_K = 1$, $\Delta_{\beta_k} = \beta_k - \beta_{k-1}$, $K > 1$

$$\text{TVO}_K^L(\theta, \phi, \mathbf{x}) := \Delta_{\beta_1} \text{ELBO}(\theta, \phi, \mathbf{x}) + \sum_{k=2}^{K} \Delta_{\beta_k} \mathbb{E}_{\pi_{\beta_{k-1}}} \left[ \log \frac{p_\theta(\mathbf{x}, \mathbf{z})}{q_\phi(\mathbf{z} \mid \mathbf{x})} \right] \le \log p(\mathbf{x}) \qquad (35)$$

$$\text{TVO}_K^U(\theta, \phi, \mathbf{x}) := \Delta_{\beta_K} \text{EUBO}(\theta, \phi, \mathbf{x}) + \sum_{k=1}^{K-1} \Delta_{\beta_k} \mathbb{E}_{\pi_{\beta_k}} \left[ \log \frac{p_\theta(\mathbf{x}, \mathbf{z})}{q_\phi(\mathbf{z} \mid \mathbf{x})} \right] \ge \log p(\mathbf{x}), \qquad (36)$$

where

$$\text{ELBO}(\theta, \phi, \mathbf{x}) := \mathbb{E}_{q_\phi(\mathbf{z} \mid \mathbf{x})} \left[ \frac{p_\theta(\mathbf{x}, \mathbf{z})}{q_\phi(\mathbf{z} \mid \mathbf{x})} \right], \qquad \text{EUBO}(\theta, \phi, \mathbf{x}) := \mathbb{E}_{p_\theta(\mathbf{z} \mid \mathbf{x})} \left[ \frac{p_\theta(\mathbf{x}, \mathbf{z})}{q_\phi(\mathbf{z} \mid \mathbf{x})} \right],$$

$$\pi_{\beta_k}(\mathbf{z}) := p_\theta(\mathbf{x}, \mathbf{z})^\beta q_\phi(\mathbf{z} \mid \mathbf{x})^{1-\beta} / Z_\beta, \qquad Z_\beta := \int p_\theta(\mathbf{x}, \mathbf{z})^\beta q_\phi(\mathbf{z} \mid \mathbf{x})^{1-\beta} \, d\mathbf{z}.$$

## D  Maximizing the TVO minimizes a divergence between the variational distribution and true posterior

We now show:

$$\text{TVO}(\theta, \phi, \mathbf{x}) = \log p_\theta(\mathbf{x}) - \mathcal{D}(q_\phi(\mathbf{z} \mid \mathbf{x}) || p_\theta(\mathbf{z} \mid \mathbf{x})) \qquad (37)$$

Where $\mathcal{D}(q_\phi(\mathbf{z} \mid \mathbf{x}) || p_\theta(\mathbf{z} \mid \mathbf{x}))$ is a divergence between the variational distribution $q_\phi(\mathbf{z} \mid \mathbf{x})$ and true posterior $p_\theta(\mathbf{z} \mid \mathbf{x})$. We refer to the notation defined in Appendix B.1 and the definition of divergence defined by Eguchi et al. [39].

*Proof.* The TVO is a left Riemann sum approximation of $\log p_\theta(\mathbf{x}) = \int_0^1 g(\beta) d\beta$, where $g(\beta) = \mathbb{E}_{\pi_\beta(\mathbf{z})}[U'(\mathbf{z})]$ and $g(\beta)$ is a differentiable monotonically non-decreasing function in $\beta$ (cf. Equation (26)). The TVO is therefore a lower bound of $\log p_\theta(\mathbf{x})$ and can be written

$$\text{TVO}(\theta, \phi, \mathbf{x}) \le \log p_\theta(\mathbf{x})$$
$$\text{TVO}(\theta, \phi, \mathbf{x}) = \log p_\theta(\mathbf{x}) - c(\theta, \phi, \mathbf{x}), \quad c(\theta, \phi, \mathbf{x}) \ge 0 \qquad (38)$$

We will show $c(\theta, \phi, \mathbf{x}) = \mathcal{D}(q_\phi(\mathbf{z} \mid \mathbf{x}) || p_\theta(\mathbf{z} \mid \mathbf{x}))$, which is equivalent to showing

① $c \ge 0, \ \forall \, p_\theta(\mathbf{z} \mid \mathbf{x}), q_\phi(\mathbf{z} \mid \mathbf{x}) \in \mathcal{Z}$

② $c = 0 \iff p_\theta(\mathbf{z} \mid \mathbf{x}) = q_\phi(\mathbf{z} \mid \mathbf{x})$

① is implied in the definition of $c$ in 38. We now show ②.

**Forward direction**  $(c = 0) \Rightarrow (p_\theta(\mathbf{z} \mid \mathbf{x}) = q_\phi(\mathbf{z} \mid \mathbf{x}))$

If $c = 0$, the left Riemann sum must be an exact approximation to $\int_0^1 g(\beta) d\beta$. Because is $g(\beta)$ is differentiable (and assuming it is finite), the Riemann approximation can only be exact when $g(\beta)$ is flat (i.e. $\frac{\partial g(\beta)}{\partial \beta} = 0$) in the region $\beta \in [0, 1]$. We first recall that by definition, $\pi_0(\mathbf{z}) = q_\phi(\mathbf{z} \mid \mathbf{x})$ and $\pi_1(\mathbf{z}) = p_\theta(\mathbf{z} \mid \mathbf{x})$. Therefore

$$\int_0^1 \frac{\partial g(\beta)}{\partial \beta} d\beta = \int_0^1 0 \, d\beta \qquad (39)$$
$$g(1) - g(0) = 0 \qquad (40)$$
$$g(1) = g(0) \qquad (41)$$
$$\mathbb{E}_{\pi_1(\mathbf{z})}[U'(\mathbf{z})] = \mathbb{E}_{\pi_0(\mathbf{z})}[U'(\mathbf{z})] \qquad (42)$$
$$\mathbb{E}_{p_\theta(\mathbf{z} \mid \mathbf{x})}[U'(\mathbf{z})] = \mathbb{E}_{q_\phi(\mathbf{z} \mid \mathbf{x})}[U'(\mathbf{z})] \qquad (43)$$

Which is only possible when $p_\theta(\mathbf{z} \mid \mathbf{x}) = q_\phi(\mathbf{z} \mid \mathbf{x})$.

**Reverse direction** $\quad \big(p_\theta(\mathbf{z}\,|\,\mathbf{x}) = q_\phi(\mathbf{z}\,|\,\mathbf{x})\big) \Rightarrow \big(c = 0\big)$

If $p_\theta(\mathbf{z}\,|\,\mathbf{x}) = q_\phi(\mathbf{z}\,|\,\mathbf{x})$, the TVO can be written as

$$\text{TVO}(\theta, \phi, \mathbf{x}) = \frac{1}{K}\sum_{k=0}^{K-1} \mathbb{E}_{\pi_{\beta_k}(\mathbf{z})}\left[\log \frac{p_\theta(\mathbf{x}, \mathbf{z})}{p_\theta(\mathbf{z}\,|\,\mathbf{x})}\right] \tag{44}$$

$$= \frac{1}{K}\sum_{k=0}^{K-1} \mathbb{E}_{\pi_{\beta_k}(\mathbf{z})}\left[\log p_\theta(\mathbf{x})\right] \tag{45}$$

$$= \log p_\theta(\mathbf{x}) \tag{46}$$

Therefore $c = 0$. $\qquad\square$

## E Derivation of the Covariance Gradient Estimator

We want to show that

$$\nabla_\lambda \mathbb{E}_{\pi_{\lambda,\beta}}[f(\mathbf{z}, \lambda)] = \mathbb{E}_{\pi_{\lambda,\beta}}[\nabla_\lambda f(\mathbf{z}, \lambda)] + \text{Cov}_{\pi_{\lambda,\beta}}\left[\nabla_\lambda \log \tilde{\pi}_{\lambda,\beta}(\mathbf{z}), f(\mathbf{z}, \lambda)\right]. \tag{47}$$

Our estimator holds under the common regularity conditions assumed for the score function estimator L'Ecuyer [40]. We begin with a simple lemma.

**Lemma 1.**

$$\nabla_\lambda \log Z_{\lambda,\beta}(\mathbf{x}) = \mathbb{E}_{\pi_\beta(\mathbf{z})}[\nabla_\lambda \log \tilde{\pi}_{\lambda,\beta}(\mathbf{z})] \tag{48}$$

*Proof of lemma 1.*

$$\nabla_\lambda \log Z_{\lambda,\beta}(\mathbf{x}) = \frac{1}{Z_{\lambda,\beta}(\mathbf{x})}\nabla_\lambda Z_{\lambda,\beta}(\mathbf{x}) \tag{49}$$

$$= \frac{1}{Z_{\lambda,\beta}(\mathbf{x})}\nabla_\lambda \int \tilde{\pi}_{\lambda,\beta}(\mathbf{z})\,\mathrm{d}\mathbf{z} \tag{50}$$

$$= \frac{1}{Z_{\lambda,\beta}(\mathbf{x})}\int \nabla_\lambda \tilde{\pi}_{\lambda,\beta}(\mathbf{z})\,\mathrm{d}\mathbf{z} \tag{51}$$

$$= \int \frac{\tilde{\pi}_{\lambda,\beta}(\mathbf{z})}{Z_{\lambda,\beta}(\mathbf{x})}\nabla_\lambda \log\tilde{\pi}_{\lambda,\beta}(\mathbf{z})\,\mathrm{d}\mathbf{z} \tag{52}$$

$$= \mathbb{E}_{\pi_\beta(\mathbf{z})}[\nabla_\lambda \log \tilde{\pi}_{\lambda,\beta}(\mathbf{z})] \tag{53}$$

$$\square$$

To prove (47), we use the product rule and rearrange

$$\nabla_\lambda \mathbb{E}_{\pi_\beta(\mathbf{z})}[f(\mathbf{z}, \lambda)] = \mathbb{E}_{\pi_\beta(\mathbf{z})}[\nabla_\lambda f(\mathbf{z}, \lambda) + f(\mathbf{z}, \lambda)\nabla_\lambda \log \pi_{\lambda,\beta}(\mathbf{z}\,|\,\mathbf{x})] \tag{54}$$

$$= \mathbb{E}_{\pi_\beta(\mathbf{z})}[\nabla_\lambda f(\mathbf{z}, \lambda) + f(\mathbf{z}, \lambda)\big(\nabla_\lambda \log \tilde{\pi}_{\lambda,\beta}(\mathbf{z}) - \nabla_\lambda \log Z_{\lambda,\beta}(\mathbf{x})\big)] \tag{55}$$

$$= \mathbb{E}_{\pi_\beta(\mathbf{z})}[\nabla_\lambda f(\mathbf{z}, \lambda)] + \mathbb{E}_{\pi_\beta(\mathbf{z})}[f(\mathbf{z}, \lambda)\nabla_\lambda \log \tilde{\pi}_{\lambda,\beta}(\mathbf{z})]$$
$$- \mathbb{E}_{\pi_\beta(\mathbf{z})}[f(\mathbf{z}, \lambda)\nabla_\lambda \log Z_{\lambda,\beta}(\mathbf{x})]. \tag{56}$$

Now using lemma 1 on the third term

$$= \mathbb{E}_{\pi_\beta(\mathbf{z})}[\nabla_\lambda f(\mathbf{z}, \lambda)] + \mathbb{E}_{\pi_\beta(\mathbf{z})}[f(\mathbf{z}, \lambda)\nabla_\lambda \log \tilde{\pi}_{\lambda,\beta}(\mathbf{z})]$$
$$- \mathbb{E}_{\pi_\beta(\mathbf{z})}[f(\mathbf{z}, \lambda)]\mathbb{E}_{\pi_\beta(\mathbf{z})}[\nabla_\lambda \log \tilde{\pi}_{\lambda,\beta}(\mathbf{z})] \tag{57}$$

$$= \mathbb{E}_{\pi_\beta(\mathbf{z})}[\nabla_\lambda f(\mathbf{z}, \lambda)] + \text{Cov}_{\pi_{\lambda,\beta}(\mathbf{z}\,|\,\mathbf{x})}\left[\nabla_\lambda \log \tilde{\pi}_{\lambda,\beta}(\mathbf{z}), f(\mathbf{z}, \lambda)\right]. \tag{58}$$

Table 1: The effect of Common Random Numbers (CRN) on TVO variance. We use the discrete model of §7.1

| Iterations | 10 | 1e6 | 2e6 | 3e6 | 4e6 |
|---|---|---|---|---|---|
| Gradient std w/o CRN | 52.33 | 2.88 | 2.57 | 2.39 | 2.47 |
| Gradient std w/ CRN | 8.19 | 1.38 | 1.17 | 1.05 | 1.03 |

# F   Variance of the Covariance Gradient Estimator and its Relationship to REINFORCE

In this section we clarify the difference between the covariance estimator (11) and the REINFORCE estimator and empirically investigate its variance.

While both estimators use the log-derivative trick, the main difference between the two is the REIN-FORCE estimator requires differentiating through $\log \pi_\beta(\mathbf{z})$ which contains the intractable normalizing constant, while the covariance estimator only requires differentiating through the unnormalized distribution $\log \tilde{\pi}_\beta(\mathbf{z})$.

We can state the difference as follows. Assuming $\pi_\beta(\mathbf{z}) = \tilde{\pi}_\beta(\mathbf{z})/Z_\beta$ depends on parameters $\lambda$, to compute $\nabla_\lambda \mathbb{E}_{\pi_\beta(\mathbf{z})}[f(\mathbf{z})]$, one can use the following gradient estimators:

$$\text{REINFORCE:} \quad \mathbb{E}_{\pi_\beta(\mathbf{z})}\left[f(\mathbf{z})\nabla_\lambda \log \pi_\beta(\mathbf{z})\right]$$

$$\text{REINFORCE + BASELINE:} \quad \mathbb{E}_{\pi_\beta(\mathbf{z})}\left[\left(f(\mathbf{z}) - \mathbb{E}_{\pi_\beta(\mathbf{z})}[f(\mathbf{z})]\right)\nabla_\lambda \log \pi_\beta(\mathbf{z})\right]$$

$$\text{COV. ESTIMATOR (ours):} \quad \mathbb{E}_{\pi_\beta(\mathbf{z})}\left[\left(f(\mathbf{z}) - \mathbb{E}_{\pi_\beta(\mathbf{z})}[f(\mathbf{z})]\right)\left(\nabla_\lambda \log \tilde{\pi}_\beta(\mathbf{z}) - \mathbb{E}_{\pi_\beta(\mathbf{z})}[\nabla_\lambda \log \tilde{\pi}_\beta(\mathbf{z})]\right)\right]$$

Unlike REINFORCE, where a baseline is typically added ad-hoc to reduce variance, the baseline $b = \mathbb{E}_{\pi_\beta(\mathbf{z})}[f(\mathbf{z})]$ naturally appears as a result of differentiating through $\pi_\beta(\mathbf{z})$ using the identity $\nabla_\lambda \log Z_{\lambda,\beta}(\mathbf{x}) = \mathbb{E}_{\pi_\beta(\mathbf{z})}[\nabla_\lambda \log \tilde{\pi}_{\lambda,\beta}(\mathbf{z})]$ derived in appendix E. The baseline also partially explains the low variance of our estimator, as it is equivalent to the "average baseline" often used reinforcement learning [41, 42].

A second source of variance-reduction comes from reusing samples, a method known as "common random numbers" [23]. The terms in the TVO are highly correlated, thus we expect reusing a single batch of samples for each additional term will act to reduce variance according to equation 8.21 in Owen [23]. However, because the covariance term breaks into both positive and negative terms, common random numbers could potentially increase variance. In Table 1 we show the average gradient std at different iterations during the training procedure, using the $S = 50$ discrete model described in §7.1 and in Figure 3 and Figure 5. It is evident reusing samples significantly reduces the variance of the covariance gradient estimator, often by more than a factor of two.

In Figure 7 we compare the variance of our estimator to the reparameterization trick and REINFORCE on the continuous model described in §7.2. To control for any possible effect on variance the additional terms in the TVO could have, we use the ELBO (i.e the TVO with $K = 1$), and plot the gradient standard deviation for the COV estimator (ours), the reparameterization trick and REINFORCE. The COV estimator has higher variance than the reparameterization trick estimator, and outperforms the REINFORCE estimator which is numerically unstable. Both the standard deviation of both the COV estimator and the reparameterization trick improves as samples increase but the effect is more prominent for the COV estimator.

# G   Special Cases of the TVO

In Table 2, we summarize the different ways the TVO generalizes existing variational objectives, and in the following subsection we list the mathematical form of each objective. In the main text, we mentioned that the lower bound variant of the TVO with $K = 1$ partition can be seen as the ELBO. This connects the TVO to all methods that maximize the ELBO. The upper bound variant of the TVO with $K = 1$ partition can be seen as EUBO. This therefore connects the TVO to all methods

Figure 7: Comparing the standard deviation of gradient estimators on continuous VAEs trained on the ELBO. The covariance estimator has higher variance than the reparameterization trick for all $S$ but much lower than REINFORCE, which is numerically unstable.

that minimize the reverse KL divergence $\text{KL}\left(p_\theta(\mathbf{z}|\mathbf{x})||q_\phi(\mathbf{z}|\mathbf{x})\right)$, including WS, RWS and inference compilation.

For $K > 1$, we have a novel objective which we can optimize with respect to $\theta$, $\phi$ or both and therefore apply to all the variational methods summarized in Table 2.

Table 2: The thermodynamic variational identity generalizes existing variational objectives.

| Approximation | | Left Riemann sum (lower bound—maximize) | | Right Riemann sum (upper bound—minimize) | |
|---|---|---|---|---|---|
| Number of partitions | | 1 | $> 1$ | 1 | $> 1$ |
| Optimize | $\theta$ | wake in WS | $\text{TVO}_K^L(\theta, \mathbf{x})$ | N/A | N/A |
| | $\phi$ | VI | $\text{TVO}_K^L(\phi, \mathbf{x})$ | wake-$\phi$ in RWS, sleep in WS, inference compilation | $\text{TVO}_K^U(\theta, \phi, \mathbf{x})$ |
| | $\theta, \phi$ | VAE | $\text{TVO}_K^L(\theta, \phi, \mathbf{x})$ | N/A | N/A |

## G.1 Variational Objective Zoo

In the following we show how a number of commonly used variational objectives can be recovered from the TVO using a single partition $K = 1$. Each method can be extended by setting $K > 1$.

We have three degrees of freedom: 1) Whether we optimize $\theta$, $\phi$, or both 2) whether we maximize $\text{TVO}_1^L(\theta, \phi, \mathbf{x})$ or minimize $\text{TVO}_1^U(\theta, \phi, \mathbf{x})$, and 3) whether we use data sampled from the true data distribution $\{\mathbf{x}_i\}_{i=1}^N \overset{\text{i.i.d}}{\sim} p(\mathbf{x})$ or from our generative model $\{\mathbf{x}_i\}_{i=1}^N \overset{\text{i.i.d}}{\sim} p_\theta(\mathbf{x})$, as in the case of inference compilation and the sleep phase of the wake-sleep algorithm.

**Variational Inference** Variational inference [26] can be recovered by learning $\phi$, maximizing $\mathrm{TVO}_1^L(\phi, \mathbf{x})$, and using real data $\{\mathbf{x}_i\}_{i=1}^N \sim p(\mathbf{x})$:

$$\phi^* = \arg\max_\phi \mathbb{E}_{x \sim p(\mathbf{x})} \left[ \mathrm{TVO}_1^L(\phi, \mathbf{x}) \right] \tag{59}$$

$$= \arg\max_\phi \mathbb{E}_{x \sim p(\mathbf{x})} \left[ \mathrm{ELBO}(\phi, \mathbf{x}) \right] \tag{60}$$

**Inference Compilation** If we instead sample data from our generative model $\{\mathbf{x}_i\}_{i=1}^N \sim p_\theta(x)$ and minimize $\mathrm{TVO}_1^U(\phi, \mathbf{x})$ we recover the inference compilation objective [27]:

$$\phi^* = \arg\min_\phi \mathbb{E}_{x \sim p_\theta(\mathbf{x})} \left[ \mathrm{TVO}_1^U(\phi, \mathbf{x}) \right] \tag{61}$$

$$= \arg\min_\phi \int p_\theta(\mathbf{x}) \left[ \mathbb{E}_{p_\theta(\mathbf{z}\,|\,\mathbf{x})} \left[ \log \frac{p(\mathbf{x}, \mathbf{z})}{q_\phi(\mathbf{z}\,|\,\mathbf{x})} \right] \right] \mathrm{d}\mathbf{x} \tag{62}$$

$$= \arg\min_\phi \int p_\theta(\mathbf{x}) \left[ \int \frac{p_\theta(\mathbf{x}, \mathbf{z})}{p_\theta(\mathbf{x})} \left[ \log \frac{p(\mathbf{x}, \mathbf{z})}{q_\phi(\mathbf{z}\,|\,\mathbf{x})} \right] \right] \mathrm{d}\mathbf{z}\, \mathrm{d}\mathbf{x} \tag{63}$$

$$= \arg\min_\phi \int \int p_\theta(\mathbf{x}, \mathbf{z}) \left[ \log \frac{p(\mathbf{x}, \mathbf{z})}{q_\phi(\mathbf{z}\,|\,\mathbf{x})} \right] \mathrm{d}\mathbf{z}\, \mathrm{d}\mathbf{x} \tag{64}$$

$$= \arg\min_\phi \mathbb{E}_{p(\mathbf{x}, \mathbf{z})} \left[ -\log q_\phi(\mathbf{z}\,|\,\mathbf{x}) \right] \tag{65}$$

**Variational Autoencoders** The loss for VAEs[1, 2] follows the same setting as in the variational inference objective, above except now we learn both $\phi$ and $\theta$.

$$\phi^* = \arg\max_{\phi,\theta} \mathbb{E}_{x \sim p(\mathbf{x})} \left[ \mathrm{TVO}_1^L(\theta, \phi, \mathbf{x}) \right] \tag{66}$$

$$= \arg\max_{\phi,\theta} \mathbb{E}_{x \sim p(\mathbf{x})} \left[ \mathrm{ELBO}(\theta, \phi, \mathbf{x}) \right] \tag{67}$$

**Wake-sleep and reweighted wake sleep** In the original wake-sleep algorithm [18], the authors proposed the *wake-phase $\theta$* update and *sleep-phase $\phi$* updates to train the generative model and inference network respectively. In Reweighted Wake-Sleep [19], two more objectives were proposed, the *reweighted wake-phase $\theta$* update[2] and the *wake-phase $\phi$* update. All except the *reweighted wake-phase $\theta$*[3] are special cases of the TVO and are listed below.

- **Wake-phase $\theta$ update** In the wake phase $\theta$ update, we consider $\phi$ fixed and maximize $\mathrm{TVO}_1^L(\theta, \mathbf{x})$, using data $\{\mathbf{x}_i\}_{i=1}^N \sim p(\mathbf{x})$ sampled from the true distribution. This is similar to the variational inference update except we're learning $\theta$ instead of $\phi$:

$$\theta^* = \arg\max_\theta \mathbb{E}_{x \sim p(\mathbf{x})}[\mathrm{TVO}_1^L(\theta, \mathbf{x})] \tag{68}$$

$$= \arg\max_\theta \mathbb{E}_{x \sim p(\mathbf{x})} \left[ \mathrm{ELBO}(\theta, \mathbf{x}) \right] \tag{69}$$

- **Sleep-phase $\phi$ update** In the sleep phase $\phi$ update, we consider $\theta$ fixed and minimize $\mathrm{TVO}_1^U(\phi, \mathbf{x})$ using simulated data $\{\mathbf{x}_i\}_{i=1}^N \sim p_\theta(\mathbf{x})$ and a single partition. This objective is the same as the inference compilation objective.

- **Wake-phase $\phi$ update** In the wake phase $\phi$ update, we instead use real data $\{\mathbf{x}_i\}_{i=1}^N \sim p(\mathbf{x})$ and again minimize $\mathrm{TVO}_1^U$:

$$\phi^* = \arg\min_\phi \mathbb{E}_{\mathbf{x} \sim p(\mathbf{x})} \left[ \mathrm{TVO}_1^U(\phi, \mathbf{x}) \right] \tag{70}$$

$$= \arg\min_\phi \mathbb{E}_{\mathbf{x} \sim p(\mathbf{x})} \left[ \mathbb{E}_{p_\theta(\mathbf{z}\,|\,\mathbf{x})} \left[ \log \frac{p_\theta(\mathbf{x}, \mathbf{z})}{q_\phi(\mathbf{z}\,|\,\mathbf{x})} \right] \right] \tag{71}$$

$$= \arg\min_\phi \mathbb{E}_{\mathbf{x} \sim p(\mathbf{x})} \left[ \mathbb{E}_{p_\theta(\mathbf{z}\,|\,\mathbf{x})} \left[ -\log q_\phi(\mathbf{z}\,|\,\mathbf{x}) \right] \right] \tag{72}$$

This is the objective given in the wake-phase $\phi$ update in equation 6 of Le et al. [20]. The gradient estimator for performing this update given in Le et al. [20] is equivalent to the gradient estimator obtained via equations (11) and (13).

# H   Additional Illustrations of the Thermodynamic Variational Identity

In Figure 8, we provide illustrations of how the $\mathbb{E}_{\pi_\beta}[U'(\mathbf{z})]$ curve relates to $\log p_\theta(\mathbf{x})$, KL $(q||p)$, KL $(p||q)$, ELBO and EUBO for the cases of ELBO $< 0 <$ EUBO and ELBO $<$ EUBO $< 0$. In the following, we provide derivations to justify the illustrations.

Figure 8: Different scenarios of the $\mathbb{E}_{\pi_\beta}[U'(\mathbf{z})]$ curve where ELBO $< 0$. On the left, $0 <$ ELBO $<$ EUBO. In the middle, ELBO $< 0 <$ EUBO. On the right ELBO $<$ EUBO $< 0$.

**Case ELBO $< 0 <$ EUBO**   The top-most point of the curve is the EUBO by definition which means that the area $A + B$ is equal to the EUBO because of the unit length of the rectangle. In a similar manner, the ELBO is the negative of the area of $C + D$. Now, due to the thermodynamic identity, $\log p_\theta(\mathbf{x}) = \int_{\beta=0}^{1} \mathbb{E}_{\pi_\beta}[U'(\mathbf{z})]\,\mathrm{d}\beta$, it is equal to $B - C$ which is the area denoted by the definite integral.

To obtain the expressions for the KL, we use the identities

$$\log p_\theta(\mathbf{x}) = \text{ELBO}(\mathbf{x}, \theta, \phi) + \text{KL}\left(q_\phi(\mathbf{z}|\mathbf{x})||p_\theta(\mathbf{z}|\mathbf{x})\right) \tag{73}$$
$$= \text{EUBO}(\mathbf{x}, \theta, \phi) - \text{KL}\left(p_\theta(\mathbf{z}|\mathbf{x})||q_\phi(\mathbf{z}|\mathbf{x})\right) \tag{74}$$

**Case ELBO $<$ EUBO $< 0$**   The top-most point of the curve is the EUBO by definition which means that $-A$ is equal to the EUBO because of the unit length of the rectangle. In a similar manner, the ELBO is $-A - B - C$. Due to the thermodynamic identity, $\log p_\theta(\mathbf{x}) = \int_{\beta=0}^{1} \mathbb{E}_{\pi_\beta}[U'(\mathbf{z})]\,\mathrm{d}\beta$, it is equal to $-A - B$ which is the area denoted by the definite integral. We obtain expressions for the KL similarly as before.

Similar line of reasoning gives rise to the relationships in Figure 8 (left).

# I   Details for Deep Generative Models

**Discrete latent variables.**   Sigmoid belief networks are used to evaluate objectives, continuous relaxations and control variate methods for learning discrete latent variable models [3–5, 19, 24, 25, 43]. The generative model is of the form $p(\mathbf{z}_{1:L}, \mathbf{x}) = p(\mathbf{z}_L) \prod_{\ell=1}^{L-1} p(\mathbf{z}_\ell|\mathbf{z}_{\ell+1}) p(\mathbf{x}|\mathbf{z}_1)$ where each conditional on $\mathbf{z}_\ell$ is an independent Bernoulli whose parameters are a linear function of $\mathbf{z}_{\ell+1}$. The likelihood $p(\mathbf{x}|\mathbf{z}_1)$ is also an independent Bernoulli whose parameters are a linear function of $\mathbf{z}_1$ and we parameterize the prior $p(\mathbf{z}_L)$.

$$p_\theta(\mathbf{z}_L) = \text{Bernoulli}(\mathbf{z}_L|\mathbf{b}_L),$$
$$p_\theta(\mathbf{z}_\ell|\mathbf{z}_{\ell+1}) = \text{Bernoulli}(\mathbf{z}_\ell|\text{decoder}_\ell(2\,\mathbf{z}_{\ell+1}-1)) \qquad \ell = L-1, \dots, 1,$$
$$p_\theta(\mathbf{x}|\mathbf{z}_1) = \text{Bernoulli}(\mathbf{x}|\text{decoder}_x(2\,\mathbf{z}_1-1) + \tilde{\mathbf{x}})$$

The inference network is factorized in the opposite way to the generative model, where $q(\mathbf{z}|\mathbf{x}) = q(\mathbf{z}_1|\mathbf{x}) \prod_{\ell=2}^{L} q(\mathbf{z}_\ell|\mathbf{z}_{\ell-1})$. Here, each conditional is an independent Bernoulli whose parameters are linear functions of the condition.

$$q_\phi(\mathbf{z}_1|\mathbf{x}) = \text{Bernoulli}\left(\mathbf{z}_1 \middle| \text{encoder}_1\left(\frac{\mathbf{x} - \bar{\mathbf{x}} + 1}{2}\right)\right),$$
$$q_\phi(\mathbf{z}_\ell|\mathbf{z}_{\ell-1}) = \text{Bernoulli}(\mathbf{z}_\ell|\text{encoder}_\ell(2\,\mathbf{z}_{\ell-1}-1)) \qquad \ell = 2, \dots, L,$$

where $\mathbf{x} \in \{0,1\}^{D_x}$ and $\mathbf{z}_\ell \in \{0,1\}^{D_z}$. We set $L = 2$, $D_x = 784$ and $D_z = 200$. We used Pytorch's default parameter initialization. The Bernoulli distributions are independent Bernoulli distributions whose parameters are logits, i.e. they get passed through a sigmoid function to obtain the probability. $\bar{\mathbf{x}}$ is the mean over training data set and $\tilde{\mathbf{x}} = \log(\bar{\mathbf{x}} - 1)$. In the linear case, the encoders and decoders are linear functions of their inputs. In the non-linear case, they are a three-layer multilayer perceptrons with $\tanh$ nonlinearities of the form $\texttt{input\_dim} \xrightarrow{\text{Lin+tanh}} D_z \xrightarrow{\text{Lin+tanh}} D_z \xrightarrow{\text{Lin}} \texttt{output\_dim}$.

We used the Adam optimizer with the learning rate in $\{3 \times 10^{-4}, 10^{-3}, 3 \times 10^{-3}\}$ and the other hyperparameters being set to the defaults. We picked the learning rate which performed best on the validation set which was $3 \times 10^{-4}$ for all algorithms. We ran the optimization for $4$ million iterations with batch size $24$.

**Continuous latent variables.** The model is of the form $p(\mathbf{z})p_\theta(\mathbf{x}|\mathbf{z}) = \text{Normal}(\mathbf{z}|0, I)\text{Bernoulli}(\mathbf{x}|\text{decoder}_\theta(\mathbf{z}))$, where $\mathbf{z}$ is 200-dimensional and $\text{decoder}_\theta$ is a three-layer multilayer perceptron with $\tanh$ activations and sigmoid output which parameterizes the probabilities of the independent Bernoulli distribution.

$$p(\mathbf{z}) = \text{Normal}(\mathbf{z}|0, I),$$
$$p_\theta(\mathbf{x}|\mathbf{z}) = \text{Bernoulli}(\mathbf{x}|\text{decoder}_\theta(\mathbf{z}))$$

The inference network is of the form $q_\phi(\mathbf{z}|\mathbf{x}) = \text{Normal}(\mathbf{z}|\text{encoder}_\phi(\mathbf{x}))$, where the encoder is a two-layer multilayer perceptron with $\tanh$ activations. The output is passed through two separate linear layers which output the mean and the log standard deviations of the independent normal distribution.

$$q_\phi(\mathbf{z}|\mathbf{x}) = \text{Normal}(\mathbf{z}|\text{encoder}_\phi(\mathbf{x})),$$

where $\mathbf{x} \in \{0,1\}^{D_x}$ and $\mathbf{z} \in \mathbb{R}^{D_z}$ for $D_x = 784$ and $D_z = 200$. The decoder is of the form $D_z \xrightarrow{\text{Lin+tanh}} D_z \xrightarrow{\text{Lin+tanh}} D_z \xrightarrow{\text{Lin}} D_x$ and its output is passed through a sigmoid to obtain probabilities for the Bernoulli distribution. The encoder is of the form $D_x \xrightarrow{\text{Lin+tanh}} D_z \xrightarrow{\text{Lin+tanh}} D_z$. Its output is passed through two *separate* neural networks of the form $D_z \xrightarrow{\text{Lin}} D_z$ which output the means and log standard deviations of the independent Normal distribution.

## J Notation

Table 3: Table of Notation

| | | |
|---:|:---:|:---|
| $\{\mathbf{x}_i\}_{i=1}^N$ | := | Data set consisting of N i.i.d samples $\mathbf{x}_i \in \mathbb{R}^D$ |
| $\{\mathbf{z}_i\}_{i=1}^N$ | := | Unobserved latent random variables $\mathbf{z}_i \in \mathbb{R}^M$ |
| $p_\theta(\mathbf{x}, \mathbf{z}) = p_\theta(\mathbf{x} \mid \mathbf{z})p_\theta(\mathbf{z})$ | := | The joint model parameterized by $\theta$, which factorizes into a likelihood $p_\theta(\mathbf{x} \mid \mathbf{z})$ and prior $p_\theta(\mathbf{z})$ |
| $p_\theta(\mathbf{z} \mid \mathbf{x}) = p_\theta(\mathbf{x}, \mathbf{z})/p_\theta(\mathbf{x})$ | := | The true (often intractable) posterior |
| $q_\phi(\mathbf{z} \mid \mathbf{x})$ | := | The variational distribution parameterized by $\phi$. By assumption $q_\phi(\mathbf{z} \mid \mathbf{x})$ is correctly normalized. |
| $\tilde{\pi}_{\lambda,\beta}(\mathbf{z}) = p_\theta(\mathbf{x}, \mathbf{z})^\beta q_\phi(\mathbf{z} \mid \mathbf{x})^{1-\beta}$ | := | The unnormalized path distributions. By construction, $\tilde{\pi}_{\lambda,\beta=1}(\mathbf{z}) = p_\theta(\mathbf{x}, \mathbf{z})$ and $\tilde{\pi}_{\lambda,\beta=0}(\mathbf{z} \mid \mathbf{x}) = q_\phi(\mathbf{z} \mid \mathbf{x})$ |
| $\pi_{\lambda,\beta}(\mathbf{z} \mid \mathbf{x}) = \tilde{\pi}_{\lambda,\beta}(\mathbf{z})/Z_{\lambda,\beta}(\mathbf{x})$ | := | The path distributions parameterized by $\lambda = \{\theta, \phi\}$ and scalar parameter $\beta \in [0, 1]$. By construction, $\pi_{\lambda,\beta=1}(\mathbf{z} \mid \mathbf{x}) = p_\theta(\mathbf{z} \mid \mathbf{x})$ and $\pi_{\lambda,\beta=0}(\mathbf{z} \mid \mathbf{x}) = q_\phi(\mathbf{z} \mid \mathbf{x})$ |
| $Z_{\lambda,\beta}(\mathbf{x}) = \int \tilde{\pi}_{\lambda,\beta}(\mathbf{z}) \, \mathrm{d}\mathbf{z}_{1:N}$ | := | The normalizing constant for the path distributions. By construction $Z_{\lambda,\beta=1}(\mathbf{x}) = p_\theta(\mathbf{x})$ and $Z_{\lambda,\beta=0}(\mathbf{x}) = 1$ (because $q_\phi(\mathbf{z} \mid \mathbf{x})$ is assumed to be correctly normalized). |
| $U_{\lambda,\beta}(\mathbf{z}) = \log \tilde{\pi}_{\lambda,\beta}(\mathbf{z})$ | := | The potential energy. |
| $U'_{\lambda,\beta}(\mathbf{z}) = \frac{\partial}{\partial\beta}U_{\lambda,\beta}(\mathbf{z})$ | := | The first derivative of the potential w.r.t $\beta$, the inverse temperature. |

## K Acronyms

**AIS** Annealed Importance Sampling

**ELBO** Evidence Lower Bound

**EUBO** Evidence Upper Bound

**IS** Importance Sampling

**IWAE** Importance Weighted Autoencoder

**KL** Kullback Leibler

**RWS** Reweighted Wake Sleep

**SGD** Stochastic Gradient Descent

**TI** Thermodynamic Integration

**TVI** Thermodynamic Variational Identity

**TVO** Thermodynamic Variational Objective

**VAE** Variational Autoencoder

**VI** Variational Inference

**VIMCO** Variational Inference For Monte Carlo Objectives

**WS** Wake Sleep

## Footnotes

[2]This was not the authors' original terminology and is used here to differentiate this objective from the original wake-phase $\theta$ update.

[3]This objective is not a special case of the TVO and is therefore not included in Table 2