[Reviews · NeurIPS 2019]

Reviewer 1



The paper connects variational inference with thermodynamic integration, so that the data log-likelihood can be formulated as a 1D integration of the instantaneous ELBO in a unit interval. By applying a left Riemann sum, TVO, a novel lower bound for the marginal log likelihood, is derived in which the traditional variational ELBO is recovered when only one partition is used. The authors then design an importance-sampling-based gradient estimator to optimize the objective, and compare with other methods on both discrete and continuous deep generative models. The paper also unifies other methods like wake sleep into the TVO framework. Originality and Significance: the formulation of TVO is an interesting idea. Better optimization methods than the importance-sampling-based approach are worth further exploring. The connections to previous methods provides a new insights about unifying different learning methods. Quality: In section 2, the development of TVO is a solid derivation. But in section 3, the authors claim the proposed method uses neither the high-variance REINFORCE nor re-parameterization. This is confusing because during the derivation, both the score-function trick and the re-parameterization trick were applied in the proposed method. During the rebuttal period, the authors provide more explanations on this both theoretically and experimentally, which I think is very beneficial. Some minor concerns: 1) f was not defined before it's used in section 3. Is it the terms multiplied by 1/K in the lhs of Eq. 2? 2) the authors provides detailed comparison with REINFORCE during the rebuttal period, and shows in math why the proposed method has lower variance, which is nice. However, the gradient of the covariance estimator in the rebuttal feedback is not very obvious to see, so it'll be good if more derivation details could be provided in the final version. 3) In the section of "The effect of S, K, and β locations", why does it "increase bias" and "are likely to be more biased" to use the importance sampler? Should it be higher variance instead? 4) As a followup of 3), is it possible that the fact limiting K=2 is enough is because the proposed method has too large variance? 5) It's great to see the authors agree to add more experiment comparisons with the results of regular ELBO optimization using REINFORCE or reparameterization. 6) It's great that the authors explain during rebuttal how the variance reduction technique in Owen (2013) can be applied in the proposed method, and provide additional experiment results to show its effect. Clarity: The organization of the submission is fine, but in terms of the overall clarity, there's still plenty of room for improvement. For example, more math background needs to be provided on WS and RWS before discussing their connections with TVO in section 4. And the readability of the second half of the paper is clearly worse than the first half. A couple descriptions about the subplot places in Figure 2 are wrong (top left should be top right, etc.). Many sentences in section 6 and 7 could be better phrased to read more like scientific writings.

Reviewer 2



UPDATE AFTER READING AUTHOR FEEDBACK ================================================================================ I would like to thank the authors for taking the concerns expressed in my review very seriously. The author feedback addresses my concerns very well. I think with the promised fixes this will be a strong paper with an original idea that could be simple enough to be used in practice. From a theory perspective, the paper might spark new ideas in readers since TI is explained very well and there are probably more connections between TI and VI. ORIGINAL REVIEW ================================================================================================== The paper proposes a series of new lower bounds to the model evidence in variational inference that generalize the standard ELBO. The proposal is based on a discretized version of thermodynamic integration (TI). Intuitively, instead of evaluating log p(x) directly by solving the intractable integral $p(x) = \int p(x,z) dz$, one first evaluates log p_0(x) for a reference model p_0(x, z) for which this is easy to do. One then changes the model p_0(x, z) on a continuous path until it is deformed to the target model p(x,z), and one integrates up the changes that this procedure incurs on the evidence log p(x). By choosing how the integral is discretized, one obtains either a lower or an upper bound on the evidence. I find the proposal convincing and well written. The underlying idea of TI is simple and well explained. A non-obvious way to obtain stochastic gradient estimates is also well explained. Experiments focus on models with discrete latent variables, as it is advertised that the proposed gradient estimator is applicable even to this situation. I am curious if the authors expect similar performance gains for models with continuous latent variables. My only main comment is that I cannot find a discussion of the variance of the gradient estimator. The proposed bound is tighter than the ELBO, i.e., it is less biased. Usually, bias reduction comes at the cost of an increase in gradient variance (see [Bamler et al., NIPS 2017] and [Rainforth et al, ICML 2018]). Larger gradient variance slows down convergence of gradient descent. The variance of the proposed gradient estimator would be interesting for another reason. The paper proposes to use a new gradient estimator although the standard REINFORCE method would in principle also work for the proposed bound. As far as I can tell, the only argument against REINFORCE gradients would be to reduce gradient variance (the use of discrete latent variables is only an argument against reparameterization gradients, not against REINFORCE). However, the proposed gradient estimator in Eq. 11 looks like it could suffer from high variance too because one has to estimate a covariance between two quantities. To estimate a covariance, one has to estimate the difference between two quantities that are typically of similar magnitude (see Eq. 12). This is hard to do numerically as absolute values cancel approximately while variances add up. The "Related Work" section mentions only work related to TI. There has been more work on tighter bounds for black box variational inference, e.g., the above mentioned papers or [Li and Turner, NIPS 2016].

Reviewer 3



*originality* The paper is very original, and the provided framework extending the standard ELBO to TVO is very elegant. I expect that this paper stimulates a lot of new research in this direction. A natural and good idea. *quality* The mathematical derivations are very clear and easy to follow. The experimental evaluation is well conducted, but restricted to MNIST only. The used implementation of TVO (based importance weighted sampling) seems to be of limited advantage (number of partitions and particles seem to have a limited impact on the learned model), which is somewhat underwhelming. Also, the effect that more particles in rws, vimco, and TVO leads to worse approximation of the posterior is surprising (Fig. 4), but not further explored. It would be interesting to see the TVI integrand (Fig. 1) for a real example/model/dataset, e.g. estimated with a massive number of importance samples. In Section 4, it is unclear to me, why both wake and sleep phases are over \phi. *clarity* The paper is largely clear. The connection to wake-sleep, however, remains somewhat unclear. *significance* While the experimental evaluation could be improved, the main contribution -- the TVO -- is very refreshing and of high significance. ************************************************************************************************************************ Update: As said in my original review, I find the proposed approach refreshing, original and creative. However, a richer set of experiments concerning datasets and models could make the paper quite stronger. Thus, I stick with my original rating (7).

Reviewer 4



The paper introduces the use of termodynamic integration for the training of variational autoencoders. The new connection between TI and the ELBO is insightful and the derivation of new lower bounds as Riemann discretization of the TI formulation of the log model evidence is clever. However, the suggested gradient estimator is quite unoriginal as it is simply a reinforce estimator with importance sampling weights. The paper is clearly written, albeit the notation is sometimes slightly confusing and some symbols are introduced before being properly defined. The experiments are good enough. Comments: 1) In line 84, the authors claim that their gradient estimator does not involve the REINFORCE estimator. However, their gradient estimation method IS the reinforce estimator with the addition of importance sampling and an extra term coming from the fact that their distribution is not normalized. 2) Eq. 2 contains several undefined symbols. It is useless and confusing to give an equation if its meaning cannot be understood in that section in which it is presented. 3) The authors claim that the performance of the method does not monotonically increases as the number of partition increases because the importance sampling scheme introduces a bias. This claim makes intuitive sense but it should be backed up by either theoretical or experimental analysis. I suggest to study a case where p(x|z) p(z) is tractable and the importance samples are unbiased.

[Author Response · NeurIPS 2019]

We thank the reviewers for their constructive feedback and address the common concerns across the four reviews.

**Difference with REINFORCE** We agree we didn't clearly explain the difference between our estimator and REIN-
FORCE. We have made the following changes to the manuscript.

**1)** Rather than stating "the gradient estimator we derive ... does not require the high-variance REINFORCE..." in the
introduction and throughout, we now say we derive a "score function estimator" to emphasize the fact that our estimator
belongs in the family of estimators that use the log-derivative trick.

**2)** We have added a new appendix section clarifying the relationship as follows. Assuming $\pi_\beta(\mathbf{z}) = \hat{\pi}_\beta(\mathbf{z})/Z_\beta$ depends
on parameters $\lambda$, to compute $\nabla_\lambda \mathbb{E}_{\pi_\beta(\mathbf{z})}[f(\mathbf{z})]$ one can use the:

$$\text{REINFORCE ESTIMATOR:} \quad \mathbb{E}_{\pi_\beta(\mathbf{z})}\left[f(\mathbf{z})\nabla_\lambda \log \pi_\beta(\mathbf{z})\right]$$

$$\text{REINFORCE ESTIMATOR + BASELINE:} \quad \mathbb{E}_{\pi_\beta(\mathbf{z})}\left[\left(f(\mathbf{z}) - \mathbb{E}_{\pi_\beta(\mathbf{z})}[f(\mathbf{z})]\right)\nabla_\lambda \log \pi_\beta(\mathbf{z})\right]$$

$$\text{COVARIANCE ESTIMATOR (ours):} \quad \mathbb{E}_{\pi_\beta(\mathbf{z})}\left[\left(f(\mathbf{z}) - \mathbb{E}_{\pi_\beta(\mathbf{z})}[f(\mathbf{z})]\right)\left(\nabla_\lambda \log \hat{\pi}_\beta(\mathbf{z}) - \mathbb{E}_{\pi_\beta(\mathbf{z})}[\nabla_\lambda \log \hat{\pi}_\beta(\mathbf{z})]\right)\right]$$

We emphasize that our estimator applies in the general case of expectations over $\pi_\beta(\mathbf{z})$, where the REINFORCE
estimator $\mathbb{E}_{\pi_\beta(\mathbf{z})}[f(\mathbf{z})\nabla_\phi \log \pi_\beta(\mathbf{z})]$ would require differentiating through $\log \pi_\beta(\mathbf{z})$ which contains the intractable
normalizing constant. Additionally unlike REINFORCE, where a baseline is typically added ad-hoc to reduce
variance, the baseline $b = \mathbb{E}_{\pi_\beta(\mathbf{z})}[f(\mathbf{z})]$ naturally appears as a result of differentiating through $\pi_\beta(\mathbf{z})$ using the identity
$\nabla_\lambda \log Z_{\lambda,\beta}(\mathbf{x}) = \mathbb{E}_{\pi_\beta(\mathbf{z})}[\nabla_\lambda \log \hat{\pi}_{\lambda,\beta}(\mathbf{z})]$ derived in appendix E.

**Low variance** To address concern 2.6 of reviewer 1 directly, in section 6.1 (Figures 3 and 4) we compare the TVO
against VIMCO which uses REINFORCE updates, and in 6.2 we compare the TVO against VAEs and IWAE which
use the reparameterization trick. In figure 4 we plot the $\phi$ gradient std of a discrete VAE and compare against VIMCO
and reweighted-wake sleep. Our method has lower variance than VIMCO which uses REINFORCE updates. However
we agree we need to discuss the low variance properties of our estimator further. We have made the following changes
to the manuscript:

**1)** After clarifying the relationship between the covariance estimator and REINFORCE in the aforementioned addition
to the appendix, we observe the $\mathbb{E}_{q_\phi(\mathbf{z})}[f(\mathbf{z})]$ baseline that is implicit in our covariance estimator is equivalent to the
"average baseline" commonly used in reinforcement learning to reduce variance. This provides a theoretical justification
for using the average baseline which is typically chosen because of empirical success and intuitive appeal.

**2)** We include additional experimental results to report the effect of using the covariance estimator to train a model on
the ELBO, and compare against the same model trained using REINFORCE and the reparameterization trick. We plot
the std. of the $\theta, \phi$ gradients across 10 trials. Preliminary results indicate the variance of our estimator is empirically
equivalent to the variance of the reparameterization trick, while the variance of the reinforce estimator is unstable
despite having 10x samples.

**3)** We include a table explicitly reporting the mean gradient std of the TVO compared to the REINFORCE-based
VIMCO (reproduced below) and update the writing in the experimental section to make it clear that section 6.1 and 6.2
are designed to compare our method against REINFORCE and the reparameterization trick respectively.

**4)** The second source of variance reduction comes from using the 'Common Random Numbers' (CRN) technique from
Owen (chapter 8.6), which we now refer to by name in the manuscript. The terms in the TVO are highly correlated,
thus we expect reusing a single batch of samples for each additional term will act to reduce variance according to
equation 8.21 in Owen. However, because the covariance term breaks into both positive and negative terms, CRN could
potentially increase variance. We therefore have included an additional experiment running the TVO with and w/o CRN
and include the results in tabular form below.

**Connections to Wake-Sleep** The endpoints of the TVI, which the TVO approximates, corresponds to the two ob-
jectives used in Wake-sleep to jointly learn a generative model and inference network. Therefore we can view the
objectives as two approximations (a left and right Riemannian sum) of a single objective, the TVI. In section 4
we discuss how the left endpoint (i.e. $\beta = 0$) corresponds to the first objective (the wake-phase $\theta$ update) which
in turn corresponds to the ELBO discussed in detail in section 1. The right endpoint (i.e. $\beta = 1$) corresponds
to the second objective, of which there are two variants (the wake-phase $\phi$ update and sleep-phase $\phi$ update) de-
termined by whether one uses real or simulated data. Revisiting the text we feel we relied too heavily on the
description of WS given by (Le et al, 2018b) and will revise the manuscript to make this connection more clear.
46

| Particles | 2 | 5 | 10 | 50 | | Iterations | 10 | 10m | 20m | 30m | 40m |
|---|---|---|---|---|---|---|---|---|---|---|---|
| Reinforce (VIMCO) | 4.48 | 8.10 | 5.72 | 5.57 | | TVO w/o CRN | 52.33 | 2.88 | 2.57 | 2.39 | 2.47 |
| TVO | 5.48 | 4.31 | 2.39 | 1.34 | | TVO w/ CRN | 8.19 | 1.38 | 1.17 | 1.05 | 1.03 |

Table 1: Mean gradient std across 10 seeds for TVO vs REINFORCE (Left) and TVO with and without common
random numbers (CRN) (Right)

[Meta-Review · NeurIPS 2019]

All reviewers agreed that the proposed method has high significance.